# *De novo* assembly of Iron-Heart *Cunninghamia lanceolata* transcriptome and EST-SSR marker development for genetic diversity analysis

**Sen Liu**[1], **Gongxiu He**[1], **Gongliang Xie**[1], **Yamei Gong**[1], **Ninghua Zhu**[1,2], **Can Xiao**[3]*

**1** Faculty of Forestry, Central South University of Forestry and Technology, Changsha, China, **2** National Long-Term Scientific Research Base for Forestry in Mid-Subtropics China, Central South University of Forestry and Technology, Changsha, China, **3** Jiangxi Environmental Engineering Vocational College, Ganzhou, China

* 17916370@qq.com

**Data Availability Statement:** All relevant data are within the paper and its Supporting Information files.

## Abstract

Iron-Heart *Cunninghamia lanceolata*, a wild relative of Chinese fir with valuable genetic and breeding traits, has been limited in genetic studies due to a lack of genomic resources and markers. In this study, we conducted transcriptome sequencing of Iron-Heart *C. lanceolata* leaves using Illumina NovaSeq 6000 and performed assembly and analysis. We obtained 45,326,576 clean reads and 115,501 unigenes. Comparative analysis in five functional databases resulted in successful annotation of 26,278 unigenes, with 6,693 unigenes annotated in all databases (5.79% of the total). UniProt and Pfam databases provided annotations for 22,673 and 18,315 unigenes, respectively. Gene Ontology analysis categorized 23,962 unigenes into three categories. KEGG database alignment annotated 10,195 unigenes, classifying them into five categories: metabolism, genetic information, biological systems, cellular processes, and environmental information processing. From the unigenes, we identified 5,645 SSRs, with dinucleotides repeats being the most common (41.47%). We observed variations in repeat numbers and base compositions, with the majority of markers ranging from 12 to 29 bp in length. We randomly selected 200 primer pairs and successfully amplified 15 pairs of polymorphic SSR primers, which effectively distinguished Chinese fir plants of different origins. This study provides insights into the genetic characteristics of Iron-Heart *C. lanceolata* and offers a foundation for future molecular marker development, breeding programs, genetic diversity analysis, and conservation strategies.

## 1. Introduction

Iron-Heart *C. lanceolata*, a native tree species and a relative of Chinese fir, is found exclusively in the Xiaoxi National Nature Reserve in Hunan province [1]. This species has a restricted habitat and a relatively small population size. Iron-Heart *C. lanceolata* exhibits remarkable wood properties, including a high heartwood ratio, hard texture, dark brown color, lustrous appearance, and resistance to decay. It surpasses fast-growing Chinese fir, red-heart Chinese fir, and common fir in terms of wood quality, making it an ideal gene bank for commercial timber

**Funding:** Our work was supported by the technology innovation project: Hunan Province Forestry Science and Technology Research and Innovation Project: XLKY2023-30, Hu Nan Forestry Bureau(XLK201921) and Study and Demonstration on Cultivation Techniques of Cunninghamia lanceolata Mixed Forest(National key R&D project No:2021YFD2201303-02.

**Competing interests:** NO authors have competing interests.

forests [2]. Despite being a valuable species, Iron-Heart *C. lanceolata* has been subject to limited research and is currently in its early stages of investigation. Studies on this species have focused on population genetics [3], seed quality [4], wood properties [2], genetic transcriptome, and molecular markers. With the international market's shift towards emphasizing both quantity and quality of wood products, the existing varieties of Chinese fir are insufficient to meet future production demands. Therefore, it is imperative to conduct comprehensive studies on Iron-Heart *C. lanceolata*. Currently, Iron-Heart *C. lanceolata* remains relatively undisturbed by human activities. However, the lack of accessibility and restricted traffic to its habitat greatly impede its protection, utilization, and overall development. It is crucial to overcome these challenges to unlock the full potential of Iron-Heart *C. lanceolata* and promote its conservation and utilization for sustainable purposes.

Simple sequence repeats (SSRs), also known as microsatellites, are short repetitive DNA motifs consisting of 1–6 base pairs that are arranged in tandem [5, 6]. These motifs can vary in the number of repeats at a specific genetic locus. SSR markers offer several advantages, including high polymorphism, strong stability, good repeatability, easy detection, simplicity of operation, and cost-effectiveness. Consequently, researchers widely utilize SSR markers in molecular-assisted breeding research, encompassing areas such as parental identification [7, 8], population genetic analysis [9, 10], fingerprint construction [11–13], and association studies of important traits [14, 15]. SSR markers can be categorized into two types based on their source: whole-genome sequences (g-SSR) and expressed sequence tag (EST-SSR) markers [16–18]. While g-SSR markers are developed from complete genomes, they tend to be expensive and laborious. On the other hand, EST-SSR markers are derived from transcriptome-expressed sequence tags, making them easier to develop for plants with incomplete genome sequences and large genomes [19–21]. Although EST-SSR markers generally exhibit lower polymorphism compared to g-SSR markers, they offer better generality and transferability, shorter development time, and direct relevance to gene function, particularly within the same genus [16, 17, 22, 23]. In recent years, the advancement of plant genome and functional genome research has resulted in extensive plant gene sequencing and the subsequent upload of numerous EST sequences to public nucleic acid databases. These EST sequences have become a valuable resource for developing EST-SSR markers [24–27]. Furthermore, with the rapid progress of high-throughput RNA sequencing technology, transcriptome sequencing provides a new avenue for studying genetic information [28, 29]. This technology allows researchers to obtain abundant EST information directly and facilitates the development of SSR molecular markers. Many studies have successfully developed EST-SSR markers based on transcriptome data in various model and non-model plants [21, 30, 31], including conifers like Korean pine [32], Masson pine [33], and Chinese fir [34]. These examples demonstrate the feasibility and convenience of utilizing plant transcriptome sequencing to obtain SSR markers. However, in the case of Chinese fir, existing markers derived from genomes or transcriptomes proved insufficient in terms of polymorphism and quantity, limiting the ability to analyze the fine spatial genetic structure of Iron-Heart *C. lanceolata* [35]. Therefore, the development of specific SSR markers is crucial to expedite marker-assisted breeding efforts for Iron-Heart *C. lanceolata*.

In this study, our primary objectives were as follows: (1) to develop dependable SSR primers based on transcriptome data and analyze the distribution of SSRs in Iron-Heart *C. lanceolata*; (2) to select polymorphic, specific, and stable SSR markers and utilize them to investigate the genetic relationships among Chinese fir varieties originating from six different regions. The findings of this study serve as a valuable reference for the ex situ conservation, fine-scale analysis of the species' spatial genetic structure, and molecular-assisted breeding of Iron-Heart *C. lanceolata* at the genomic level.

**Table 1. Information on Chinese fir plants of six different origins.**

| Origin | Species | Altitude | Location | MAP (mm) | Abbreviation |
|---|---|---|---|---|---|
| Hunan, Yongshun | Iron-Heart *C. lanceolata* | 849 m | E 110°15′, N 28°48′ | 1357 mm | HN-YS |
| Fujian, Shunchang | *C. lanceolata*-020 | 1295 m | E 117°45′, N 27°10′ | 1688 mm | FJ-020 |
| Fujian, Shunchang | *C. lanceolata*-061 | 1383 m | E 117°45′, N 27°10′ | 1688 mm | FJ-061 |
| GuangXi, Chenshan | Red-heart Chinese fir | 1135 m | E 114°35′, N 27°20′ | 1663 mm | GX-CS |
| Hunan, Zhangjiajie | Fast-growing Chinese fir | 265 m | E 110°40′, N 29°20′ | 1973 mm | HN-ZJJ |
| Hunan, Youxian | Chinese fir | 143 m | E 27°01′, N 113°15′ | 1410 mm | HN-YX |

## 2. Materials and methods

### 2.1. Plant materials

Seeds were collected from the Hunan Xiaoxi National Nature Reserve in 2019 and subsequently planted in the Botanical Garden of Central South University of Forestry and Technology in March 2020. In December 2020, leaves from the Iron-Heart *C. lanceolata* seedlings were carefully collected, wrapped in tin foil, and quick-frozen using liquid nitrogen. Transcriptome sequencing was conducted by Igenebook (Wuhan, China). For the selection and application of SSR primers, materials were obtained from Chinese fir plants representing six different origins, which were sourced from the Chaling Chinese fir germplasm resource collection nursery in Hunan. A total of 5–6 plants were collected from each origin, and detailed information and abbreviation are provided in Table 1.

### 2.2. Sequencing and annotation

After promptly grinding the leaves of Iron-Heart *C. lanceolata* in liquid nitrogen until they turned into a fine powder, we proceeded with RNA extraction. We meticulously assessed the purity and integrity of the extracted RNA before proceeding to construct a sequencing library. The high-throughput sequencing was performed using the state-of-the-art Illumina NovaSeq 6000 platform. To ensure data quality, we utilized FastQC to evaluate the original reads, swiftly removing reads with adapters, N reads exceeding 10%, and low-quality reads. As a result, we obtained a set of clean reads suitable for further analysis. To assemble the reads and obtain transcript fragments, we employed the Trinity software [36]. Additionally, hierarchical clustering was performed using Corset [37] to obtain a comprehensive set of nonredundant unigenes. To enhance the functional understanding of our assembled sequences, we carried out alignment and annotation procedures using five essential databases. These databases include Protein Families (Pfam) (http://pfam.sanger.ac.uk/), Universal Protein (UniProt) (https://www.uniprot.org/), Gene Ontology (GO) (http://www.geneontology.org), Kyoto Encyclopedia of Genes and Genomes (KEGG) (https://www.genome.jp/kegg/), and COG (http://www.ncbi.nlm.nih.gov/COG/). By utilizing these comprehensive databases, we aimed to elucidate the functions and pathways associated with the assembled sequences, thereby gaining deeper insights into the molecular characteristics of Iron-Heart *C. lanceolata*.

### 2.3. SSR design

We employed the MISA software (http://pgrc.ipk-gatersleben.de/misa/) to identify SSR markers within the unigene sequences, with a minimum repeat sequence length of 18 bp. Following the identification of SSR markers, we utilized Primer 6.0 software (Premier Biosoft International, Palo Alto, CA, USA) for primer design. The primer length ranged from 18 to 25 bp, ensuring optimal specificity. The melting temperature (Tm) value of the primers fell within

the range of 52.0˚C to 60.0˚C, with a maximum difference of 5˚C between the Tm values of the upstream and downstream primers. The (G + C) content of the primers was maintained between 40% and 60% to ensure stability and efficient amplification. Furthermore, the primer amplification length was designed to be within the range of 100 bp to 300 bp, providing a suitable target size for PCR amplification.

## 2.4. EST-SSR Screening and relationship identification of Chinese fir

We conducted a primer screening by randomly selecting 200 SSRs, using a panel of six Iron-Heart *C. lanceolata* samples. To verify the polymorphism of the primers, Chinese fir plants from six different origins were utilized. The PCR amplifications were carried out in a 20 μl reaction volume, consisting of 4 μl of template DNA, 10 μl of 1 × Tap PCR Mix (Tiangen) DNA, 1.0 μl of each primer, and 4 μl of sterile distilled water. The amplification was performed on an Applied Biosystems 9700 thermocycler, employing a touchdown protocol. Initially, a denaturation step was conducted at 95˚C for 5 min, followed by 35 cycles of denaturation at 94˚C for 30 s, annealing at 65–55˚C and 72˚C for 30 s, and extension at 60˚C for 10 min. The final step involved storing the samples at 4˚C. The PCR products were separated on an 8% polyacrylamide gel and visualized using silver nitrate staining, following established protocols [38]. The primer synthesis and PCR product detection were conducted by Sangon in Shanghai, a trusted service provider for these procedures in our study.

## 2.5. Statistical analysis

We used GeneMarker 2.20 software [39] to analyze the genotyping results. In GenALEx 6.5 [40], we examined the number of alleles ($N_a$), observed heterozygosity ($H_O$), expected heterozygosity ($H_e$), and constructed a genetic distance matrix between the samples. For the cluster analysis based on genetic distance, we employed MEGA software [41] and applied the unweighted pair group method with arithmetic mean (UPGMA). To enhance the visual presentation of the cluster analysis structure diagram, we used the online software ITOL [42] for its creation and editing. This step allowed us to produce aesthetically pleasing graphics that effectively represented the results of the cluster analysis.

## 3. Results

### 3.1. Transcriptome data assembly and unigene annotation

A total of 45,326,576 clean reads were obtained from the transcriptome data of Iron-Heart *C. lanceolata*. The average GC content of the reads was 44.94%, and the Q30 value was 93.62%, indicating high sequencing quality and suitability for further analysis. Through splicing with Trinity, we obtained 184,918 transcripts, resulting in 115,501 unigenes with an average length of 654.62 bp. The assembly quality was reflected in the lengths of N10-N50, which were 4190 bp, 3049 bp, 2351 bp, 1800 bp, and 1258 bp, respectively. The assembly data demonstrated a satisfactory quality level (Table 2).

The gene function annotation of the 115,501 unigenes was successfully conducted in five databases. Among them, 6,752 unigenes were annotated in all five databases, representing 5.85% of the total, while 27536 unigenes were annotated in at least one database, accounting for 23.84% of the total (Fig 1). Notably, the GO database yielded the highest number of successful annotations, with 23,962 unigenes annotated, constituting 20.75% of the total. Conversely, only 8.82% of the unigenes were successfully annotated in the KEGG database.

**Table 2. Distribution characteristics of Iron-Heart *C. lanceolata* transcriptome.**

| Item | Number or Length |
|---|---|
| Raw reads | 45,422,614 |
| Clean reads | 45,326,576 |
| Transcripts | 184,918 |
| Unigenes | 115,501 |
| N10 | 4,190 bp |
| N20 | 3,049 bp |
| N30 | 2,351 bp |
| N40 | 1,800 bp |
| N50 | 1,258 bp |
| Average unigene | 654.62 bp |
| GC content | 44.94% |
| Q30 | 93.62% |

## 3.2. Gene Ontology (GO) enrichment and Kyoto Encyclopedia of Genes and Genomes KEGG pathway analysis of unigenes

We conducted the GO functional classification of the unigenes using Blast2GO software [43]. A total of 23,963 unigenes were successfully annotated in the GO database, and they were categorized into three main functional categories: biological processes, cellular components, and molecular functions. Additionally, these unigenes were further classified into 30 subcategories (Figs 1 and 2A). The annotated unigenes exhibited diverse functions, including involvement in metabolic processes and cellular processes in the biological process category, cellular components and cell-related functions in the cellular component category, and various molecular functions such as ATP binding, RNA binding, zinc ion binding, and endonuclease activity in the molecular function category.

To gain deeper insights into the biological function of Iron-Heart *C. lanceolata*, we conducted a comprehensive comparison of all the unigenes against the KEGG and Pathway databases. Among the results, a total of 10,196 unigenes (14.33%) showed significant matches and were classified into five major categories. The largest category was the BRITE hierarchies, with three subcategories encompassing a total of 8,001 annotated unigenes. Following this, we observed significant representation in the metabolism, genetic information processing, cellular processes, and environmental information processing categories. The KEGG functional analysis emerged as a valuable resource for exploring specific processes, pathways, and molecular functions associated with the Iron-Heart *C. lanceolata* transcriptome (Fig 2B).

## 3.3. SSR identification in Iron-Heart *C. lanceolata*

We conducted a comprehensive search for SSR loci within the 115,501 unigenes, resulting in the identification of 5,645 SSRs. The SSRs exhibited a distribution frequency of 4.88%, with an average length of 14.28 bp and an average SSR distribution distance of 4.35/kb. The SSRs demonstrated a diverse array of repeat types, as indicated by their abundance (Table 3). Among the identified SSRs, trinucleotides repeats constituted the majority, accounting for 52.38% of the total, followed by dinucleotides repeats at 41.47%, and tetranucleotides repeats, which were the least frequent, accounting for 0.62% of the total. Overall, the 5,645 SSRs encompassed a total of 186 repeat motif types, with each nucleotide exhibiting distinct repeat motif compositions (Table 3). Notably, the AT/TA repeat motif displayed the highest frequency (960 times), slightly surpassing the AG/CT motif (873 times). Among the trinucleotide repeats, the AAG/

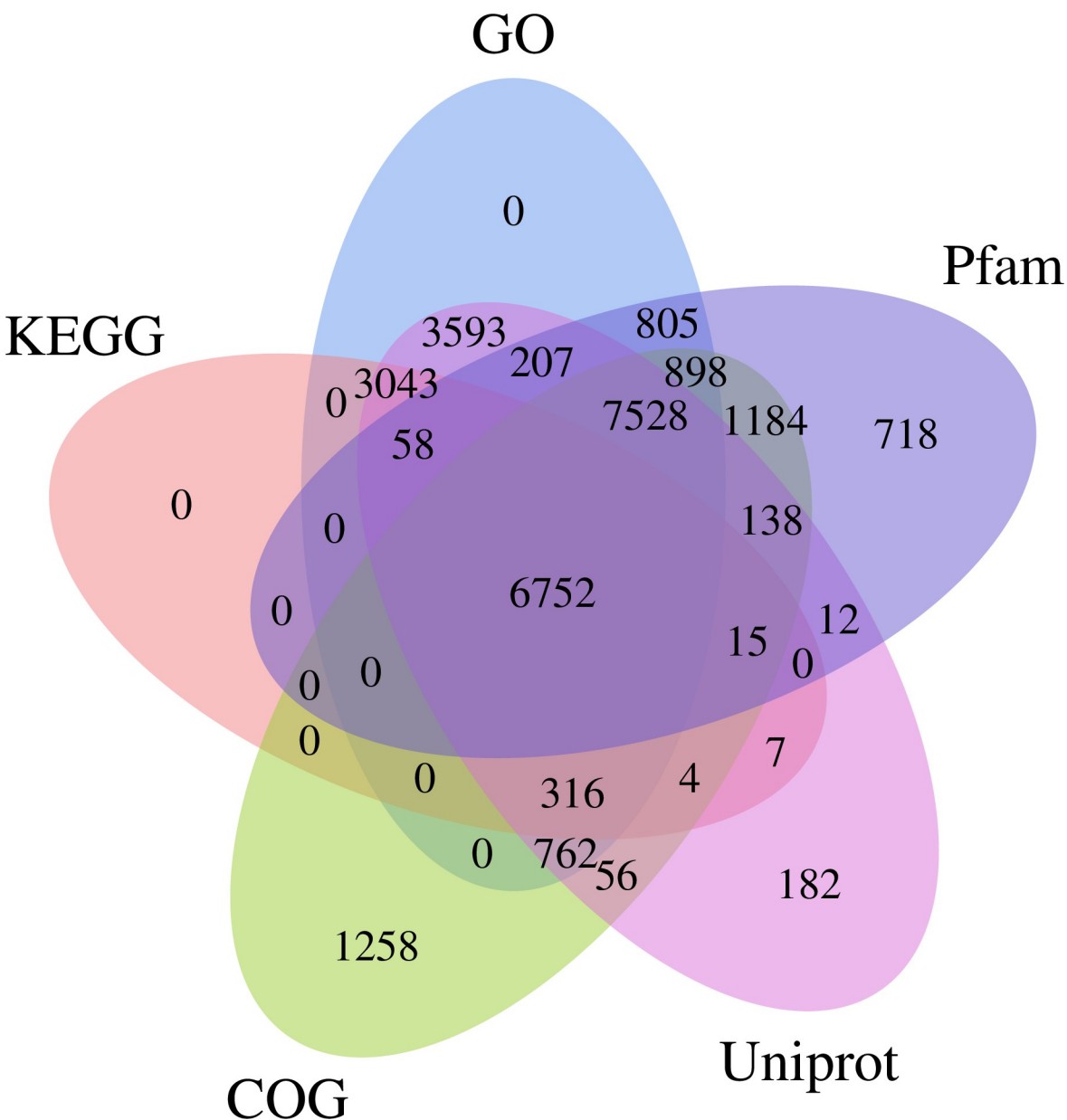

**Fig 1. Venn diagram of unigene annotations against Uniprot, Pfam, KEGG, GO, and COG databases.**

CTT motif was the most prevalent (740 times), with nearly double the number of repeats compared to the AGG/CCT repeat motif (474 times). Regarding the TTNR-HXNR repeats, the primary repeat motif types and their respective frequencies were as follows: AAAT/ATTT (35), AAGAG/CTCTT (10), and AAGAGG/CCTCTT (20). As shown in Fig 3., a clear motif type bias existed in Iron-Heart *C. lanceolata*. The frequency of the AT/TA motif (17.00%) was much higher than other motifs, followed by AG/CT (15.46%), AAG/CTT (13.11%), and AC/GT (8.98%).

When considering the occurrence frequency and distribution density of each repeat type, the ranking from smallest to largest was as follows: pentanucleotides, hexanucleotides, tetranucleotides, dinucleotides, and trinucleotides. Similarly, the average distance of each repeat type

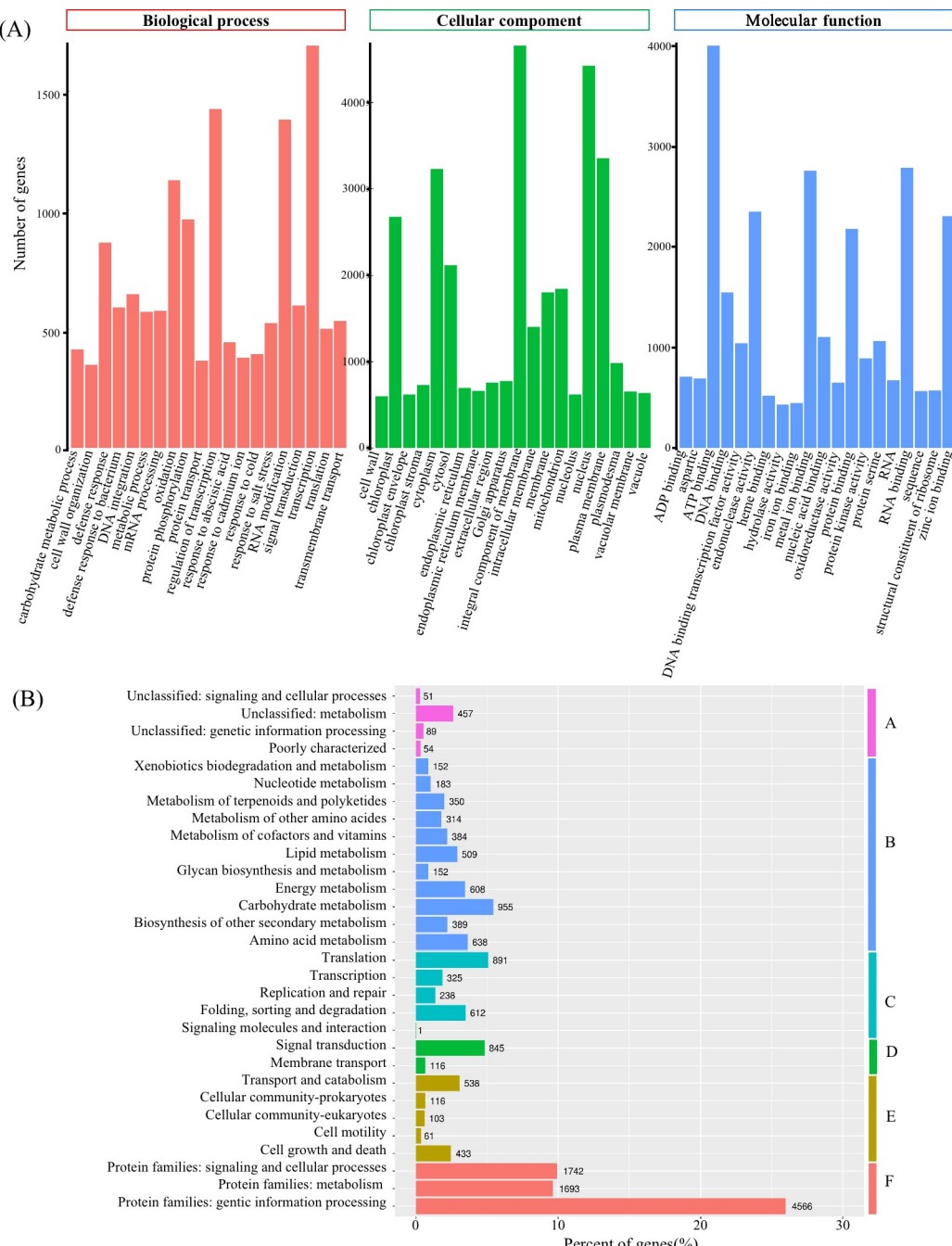

**Fig 2.** GO annotations of unigenes and clusters of orthologous groups based on KEGG classification: **A:**not included in Pathway or BRITE; **B:** metabolism; **C:**genetic information processing; **D:** environmental information processing; **E:** cellular processes; **F:**BRITE hierarchies.

followed a ranking from smallest to largest as: trinucleotides, dinucleotides, tetranucleotides, hexanucleotides, and pentanucleotides. In general, the distribution density exhibited an increasing trend with the number of SSR loci, whereas the average distance displayed the opposite trend.

**Table 3. Distribution of SSR motifs in Iron-Heart *C. lanceolata* transcriptome.**

| Repeat Motif | SSR Number | Proportion (%) | Frequency (%) | Average Distance (kb) | Distribution Density | Average Length | Repeat Type | Main Repeat Type |
|---|---|---|---|---|---|---|---|---|
| DNRs | 2341 | 41.47 | 2.03 | 32.30 | 30.96 | 16.70 | 11 | AT/TA (960) |
| TNRs | 2957 | 52.38 | 2.56 | 25.57 | 39.11 | 17.27 | 59 | AAG/TTC (740), AGG/CCT (474) |
| TTNRs | 180 | 3.19 | 0.16 | 420.05 | 2.38 | 23.87 | 49 | AAAT/ATTT (35) |
| TTNRs | 35 | 0.62 | 0.03 | 2160.26 | 0.46 | 25.86 | 17 | AAGAG/CTCTT (10) |
| HXNRs | 132 | 2.34 | 0.11 | 572.80 | 1.75 | 34.23 | 50 | AAGAGG/CCTCTT (20) |
| Total | 5645 | 100.00 | 15.03 | 4.35 | 229.63 | 14.28 | 186 | |

Note: DNRs: dinucleotides; TNRs: trinucleotides; TTNRs: tetranucleotides; RTNR: pentanucleotides; HXNRs: hexanucleotides.

## 3.4. Analysis of repeat motif

We conducted an analysis of the frequencies of EST-SSRs with different numbers of tandem repeats, and the results are depicted in Fig 3A. Notably, we observed substantial variations in the repetition numbers among different SSR repeat types, resulting in distinct types of loci. The repetition numbers ranged from 5 to 23 (Table 4). Dinucleotide repeats exhibited repetition numbers ranging from 6 to 23, while trinucleotide repeats showed relatively larger repetition numbers, ranging from 5 to 21. Regarding four, five, and six nucleotide repeats, the most common repetition number was 5. Overall, SSRs with 5 to 10 tandem repeats per locus were the most prevalent, accounting for 90.01% of the total SSRs, followed by 11 to 15 tandem repeats, which accounted for 6.86%. The remaining repetition numbers constituted less than 3.14% of the total. The general trend observed was that as the number of repetitions increased, the frequency of occurrence decreased (S1 Table).

Furthermore, we conducted a detailed analysis of the nucleotide repeats associated with SSR lengths (S2 Table). The sizes of all nucleotide repeats ranged from 12 to 83 bp, with each repeating base varying from 12 to 75 bp, 15 to 63 bp, 20 to 80 bp, 25 to 55 bp, and 30 to 54 bp,

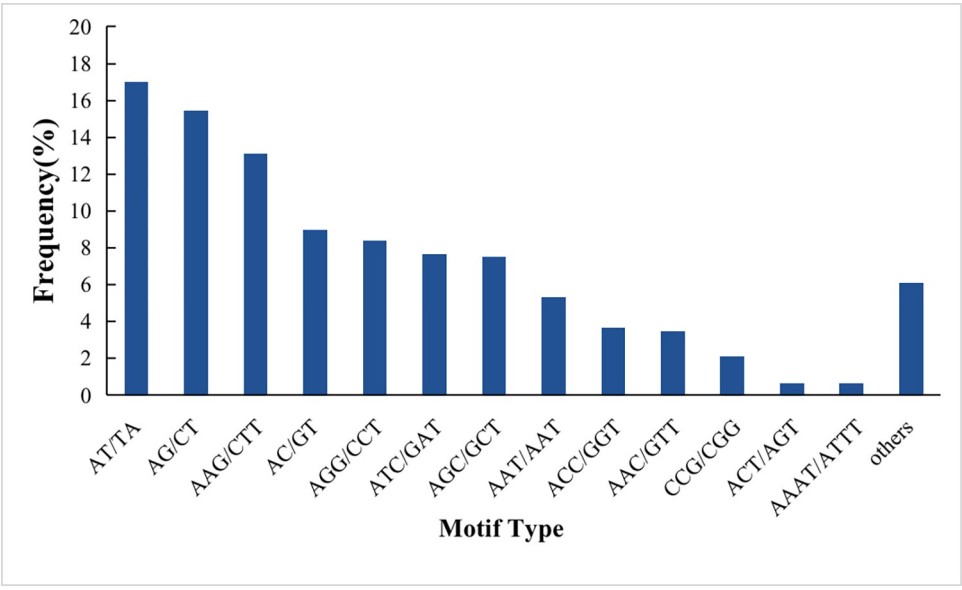

**Fig 3. Frequency distribution of EST-SSRs based on motif types.**

**Table 4. Summary of EST-SSRs identified in Iron-Heart *C. lanceolata* transcriptome.**

| Number of repeat unit | Motif length | | | | | | |
|---|---|---|---|---|---|---|---|
| | DNRs | TNRs | TTNRs | RTNRS | HXNRs | Total | % |
| 5 | 0 | 1751 | 118 | 34 | 70 | 1973 | 34.95% |
| 6–7 | 1471 | 950 | 46 | 0 | 54 | 2521 | 44.66% |
| 8–9 | 378 | 194 | 7 | 0 | 8 | 587 | 10.40% |
| 10–11 | 121 | 44 | 0 | 1 | 0 | 166 | 2.94% |
| 12–13 | 125 | 14 | 2 | 0 | 0 | 141 | 2.50% |
| 14–15 | 76 | 2 | 2 | 0 | 0 | 80 | 1.42% |
| 16–17 | 60 | 0 | 2 | 0 | 0 | 62 | 1.10% |
| 18–19 | 31 | 1 | 2 | 0 | 0 | 34 | 0.60% |
| 20–21 | 39 | 1 | 1 | 0 | 0 | 41 | 0.73% |
| 22–23 | 28 | 0 | 0 | 0 | 0 | 28 | 0.50% |
| > = 24 | 12 | 0 | 0 | 0 | 0 | 12 | 0.21% |
| Total | 2341 | 2957 | 180 | 35 | 132 | | |
| % | 41.47% | 52.38% | 3.19% | 0.62% | 2.34% | | |

respectively. Only 3.84% of the total SSR lengths were equal to or greater than 30 bp, while the remaining 96.16% of SSR fragments fell within the range of 10 to 29 bp in length. Specifically, dinucleotide repeats were approximately 12 bp in size, constituting 6.23% of the total nucleotides and 46.18% of the dinucleotide repeats. Trinucleotide repeats were predominantly 15 bp in length (approximately 1,751 occurrences), representing 10.09% of the total nucleotides. The lengths of 20 bp, 25 bp, and 30 bp accounted for proportions below 1% of the total nucleotides. The length of SSRs may affect their evolution or have functional significance for genes in physiology and development. In Iron-Heart *C. lanceolata*, 21.17% of SSRs were categorized as Class I microsatellites and 78.83% as Class II microsatellites.

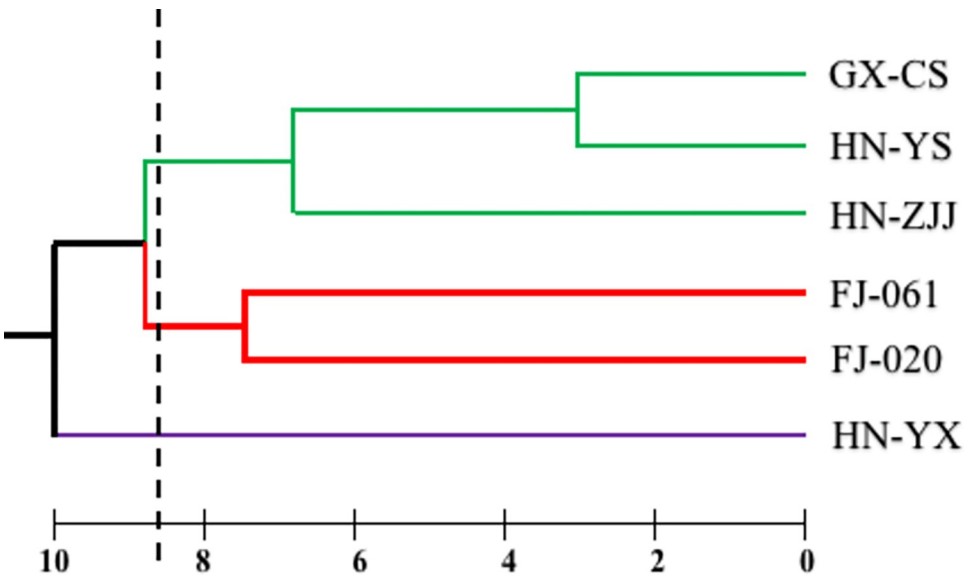

**Fig 4. Hierarchical clustering analysis using UPGMA based on 15 EST-SSR markers of 32 samples.**

### 3.5. EST-SSR versatility detection and application

To assess the reliability and cross-species transferability of the selected primers, we conducted PCR amplification and primer screening using Iron-Heart *C. lanceolata* DNA. Out of the initial pool, we chose 15 SSRs for fluorescence labeling and examined polymorphisms in 32 Chinese fir plants from six different origins. Surprisingly, the amplification success rate was only 7.5%. Comprehensive information on the 15 SSRs is provided in S2 Table. We utilized IHSSR03 to amplify four capillary electrophoresis templates, as illustrated in S1 Fig. Through this analysis, we identified a total of 51 alleles using the 15 SSR primer pairs. The number of effective alleles ranged from 1.128 to 2.851. Additionally, the average observed heterozygosity and average expected heterozygosity were calculated as 0.301 and 0.355, respectively (S3 Table). The cluster analysis results enabled the classification of the 32 samples into three main groups, as depicted in Fig 4 (the abbreviation details were provided in Table 1). Notably, the genetic distance between Iron-Heart *C. lanceolata* and red-heart Chinese fir was the smallest, indicating their close relationship. Furthermore, by consolidating the samples from Fujian, we successfully differentiated them from other Chinese fir plants of diverse origins using the SSR markers developed from the Iron-Heart *C. lanceolata* transcriptome.

## 4. Discussion

### 4.1. Unigene assembly and annotation

Transcriptome sequencing is one of the powerful tools for investigating the gene expression profiles and functional characteristics of biological tissues or cells. By sequencing the coding sequences (CDS), researchers have explored differential gene expression, regulatory mechanisms, functional gene discovery, and marker development, such as SSRs and SNPs [28, 44–46]. SSR markers developed from transcriptome sequencing, known as EST-SSR markers, are particularly valuable as they are closely associated with functional genes [46–48]. These markers directly reflect transcriptomic differences without the need for library construction or screening. In our study, annual leaves of Iron-Heart *C. lanceolata* were sequenced through high-throughput transcriptome, and SSR markers were developed. In our study, a total of 45,422,614 raw reads were obtained, which were subjected to stringent quality control and data filtering, resulting in 45,326,576 clean reads. The Q30 value, exceeding 80%, reached an impressive 93.62%, indicating the accuracy and reliability of the sequencing data. The N50 value was 1258 bp, and the GC content was 44.94%. The de-novo transcriptome assembly yielded satisfactory results, indicating its usability. Overall, 115,501 unigenes were successfully obtained from the transcriptomic data. To gain insights into the biological significance of these unigenes, sequence alignments and gene function annotations were performed using five databases. These annotated sequences lay a foundation for further studies of genetic diferentiation in Iron-Heart *C. lanceolata*. GO analysis revealed successful matching of 23,963 unigenes, which were classified into three functional categories: biological processes, cellular components, and molecular functions, encompassing 20 subcategories. Comparison and analysis of the KEGG and Pathway databases resulted in the annotation of 10,196 unigenes, spanning BRITE hierarchies, cellular processes, environmental information processing, genetic information processing, and metabolism.

### 4.2. Marker discovery

EST-SSR markers are a useful tool for analyzing genetic structures and fine spatial genetic structures of species, creating fingerprint maps, and identifying the male parents of offspring [9, 10]. However, there are currently no SSR markers available for Iron-Heart *C. lanceolata*,

which greatly limits the work of molecular-assisted breeding and ex situ conservation. In this study, a total of 5,645 polynucleotide repeat motifs from 115,501 unigenes of Iron-Heart *C. lanceolata* were discovered. The SSR frequency was 4.88%, which is comparable to that of Korean pine [21], Masson pine [33], *Taxus cuspidata* [49], *Pinus elliottii Engelm* [50], and red-heart Chinese fir [51]; however, it was lower than the SSR distribution frequencies of peony [31], ginger [52], mung bean [53], and other crops. The size of the database, the SSR site search software and conditions, and different organizations all have impacts on the frequency of the SSR distribution [54]. Based on the Iron-Heart *C. lanceolata* transcriptome data, we obtained SSR markers with six types of repeat motifs. The ratios of nucleotide repeats were quite different, with single nucleotides being the most common at 67.49%, which is consistent with the results of Chen Xingbin [51]. Differences in the distribution of SSR sequences in different species may be related to differences in genome size between species. Differences in genome size and base ratio cause substantial differences in the distributions of dominant SSR sequences. The `AT/TA` motif was the main dominant dinucleotide repeat motif of Iron-Heart *C. lanceolata*, accounting for 5.53% of the total number of SSRs, while the `CG/CG` motif appeared only once in this study. The `AAG/CCT` and `AGG/CTT` motifs were the main repeating motifs of the trinucleotides, while the `ATC/GAT` motif also appeared at a higher frequency in this study, accounting for 2.49% of the total SSRs, which was similar to the results of a study on the precious material *Michelia macclurei* [55]. We also found 118 `CCGs/CCGs` in the EST sequence of Iron-Heart *C. lanceolate* (S2 Table). This phenomenon was substantially more pronounced in the monocotyledonous plants than in the dicotyledonous plants, and its content was higher than that in ramie [56]. Their existence may be related to specific functions, such as stress resistance, cold resistance, or signaling and transduction; however, we require further related research for verification.

The dominant dinucleotide repeat motif in Iron-Heart *C. lanceolata* was found to be `AT/TA`, accounting for 5.53% of the total number of SSRs. Interestingly, the `CG/CG` motif was observed only once in this study. Regarding trinucleotide repeats, the `AAG/CCT` and `AGG/CTT` motifs were identified as the main repeating motifs. Additionally, the `ATC/GAT` motif was found to occur at a relatively higher frequency, accounting for 2.49% of the total SSRs, which aligns with the findings from a study on *Michelia macclurei* [55], an important plant species. Notably, our analysis also revealed the presence of 118 `CCGs/CCGs` in the EST sequence of Iron-Heart *C. lanceolata*. This observation is more prominent in monocotyledonous plants compared to dicotyledonous plants and shows higher content compared to ramie. These specific motifs may possess functional significance, such as involvement in stress resistance, cold resistance, signaling, and transduction pathways. However, further research is required to validate and explore their specific roles and mechanisms in Iron-Heart *C. lanceolata*.

## 4.3. Causes of SSR polymorphisms

The presence of different repeat types and repeat lengths contributes significantly to the high sequence polymorphism observed in SSR markers. It has been observed that the number of SSR alleles tends to increase with an increase in the number of core sequence repeats, indicating a positive correlation [57]. In our study, we observed a decrease in the abundance of EST-SSRs with an increase in the number of repeat types. Additionally, within the same repeat nucleotide sequence, the occurrence of SSRs decreased as the number of repeats increased. This variation in SSRs of different lengths provides opportunities for the development of highly polymorphic SSR markers.During the development of SSR primers, we excluded single nucleotide repeats due to their susceptibility to mismatches. The remaining SSR repeats were

predominantly observed between 5 and 13 repeats, with some instances of even higher repeats, reaching up to 25 repeats. In terms of SSR fragment length, our analysis revealed that the majority (88.96%) of the SSR fragments were less than 20 bp in length. Furthermore, a significant proportion (25.64%) of the sequences comprised 2–6 nucleotide repeats, reflecting the diversity in SSR lengths.These findings highlight the dynamic nature of SSR markers, their association with repeat types and lengths, and the potential for developing highly polymorphic SSR markers of varying lengths, thereby facilitating genetic studies and breeding programs in Iron-Heart *C. lanceolata*. Previous studies have suggested that the characteristically short lengths of SSRs may have functional implications with respect to their evolution or the genes involved in plant physiology and development. Tree peony SSRs were divided into two groups, 85% of SSRs were categorized as Class I microsatellites and 1% as Class II microsatellites. In our study, 21.17% of SSRs were categorized as Class I microsatellites and 78.83% as Class II microsatellites, and the proportion of Class I is higher than tree peony's [58].

## 4.4. Cross-species transferability of SSR markers and relationship identification of Chinese fir plants of six different origins

To assess the general applicability and polymorphism of the SSR markers developed from the Iron-Heart *C. lanceolata* transcriptome, we randomly selected 200 markers and identified 15 markers that were stable, specific, and exhibited polymorphism. These selected markers were further utilized to investigate the genetic relationships among Chinese fir plants from six different origins. The observed heterozygosity and expected heterozygosity values for the 15 SSR markers in the 32 samples were determined to be 0.301 and 0.335, respectively. These values are consistent with the findings reported in a study involving Chinese fir plants from 12 different origins [59], but notably lower than those reported in Duan Hongjing's study [60]. The observed variations in heterozygosity may be attributed to factors such as the number of markers and populations analyzed, population structures, the size of samples and interpopulation affinities. Based on the UPGMA analysis of the 15 SSR markers, the samples were classified into three distinct groups. Notably, the phylogenetic relationship between red-heart Chinese fir and Iron-Heart *C. lanceolata* appeared relatively close, indicating a potential genetic association. However, further investigation is required to ascertain the significance and underlying factors contributing to this relationship. Previous research has indicated comparable wood densities between these species [2], but the specific differences between them warrant further study. These findings demonstrate the utility of the selected SSR markers for evaluating genetic relationships and provide valuable insights into the genetic diversity and population structures of Chinese fir plants. Further research is essential to unravel the intricacies of the observed phylogenetic patterns and to explore the potential implications of the identified genetic associations in relation to wood characteristics and other important traits.

## 5. Conclusion

We successfully annotated a total of 26,278 out of 115,501 unigenes using five comprehensive databases. Through this annotation, we identified a significant number of EST-SSRs, totaling 5,645. From this pool, we randomly selected 200 SSR primers and meticulously screened them, resulting in the identification of 15 pairs of highly polymorphic primers. Subsequently, we employed these markers to investigate the genetic relationships among Chinese fir varieties originating from different regions.The clustering analysis using the 15 SSR markers demonstrated their efficacy in effectively distinguishing Chinese fir varieties of different origins. Notably, the relative genetic relationship between red-heart Chinese fir and Iron-Heart *C. lanceolata* was found to be the closest. However, further investigations incorporating phenotyping

and molecular approaches are necessary to comprehensively understand the differences between these two varieties.To the best of our knowledge, although SSR markers have been previously developed based on the Chinese fir transcriptome, our study represents the first attempt to leverage transcriptome databases to develop a comprehensive set of EST-SSR markers specifically for Iron-Heart *C. lanceolata*. The results of our research provide a solid foundation for conducting analyses on the fine spatial structure, population genetic structure, and molecular-marker-assisted breeding of Iron-Heart *C. lanceolata*. Overall, our findings contribute to the existing knowledge in this field and pave the way for future studies aimed at elucidating the genetic characteristics and practical applications of Iron-Heart *C. lanceolata*.

## Supporting information

**S1 Table. The percentage of SSR repeat motif type in Iron-Heart *C.lanceolata* transcriptome.**
(XLSX)

**S2 Table. The number of the different motif length in 5 motif type in Iron-Heart *C.lanceolata* transcriptome.**
(XLSX)

**S3 Table. The information of the 15 EST-SSRs and the genetic parameters in 32 samples.**
(XLSX)

**S1 Fig. Amplification results of SSR IHSSR03 in 4 samples.**
(TIF)

**S1 Data.**
(XLSX)

**S2 Data.**
(XLSX)

## Author Contributions

**Conceptualization:** Gongxiu He, Ninghua Zhu, Can Xiao.

**Data curation:** Sen Liu.

**Formal analysis:** Sen Liu, Gongliang Xie.

**Funding acquisition:** Ninghua Zhu, Can Xiao.

**Investigation:** Sen Liu, Gongliang Xie, Yamei Gong.

**Visualization:** Ninghua Zhu.

**Writing – original draft:** Sen Liu, Can Xiao.

**Writing – review & editing:** Sen Liu, Ninghua Zhu, Can Xiao.

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
