## [Decision Letter · Decision Letter 0]

6 Mar 2023

PONE-D-22-35149De novo assembly of Iron-Heart Cunninghamia lanceolat a  transcriptome using Illumina sequencing and EST-SSR marker development for genetic diversity analysis of Chinese firPLOS ONE

Dear Dr. Xiao,

Thank you for submitting your manuscript to PLOS ONE. After careful consideration, we feel that it has merit but does not fully meet PLOS ONE’s publication criteria as it currently stands. Therefore, we invite you to submit a revised version of the manuscript that addresses the points raised during the review process.

We look forward to receiving your revised manuscript.

Kind regards,

Pramod Prasad, Ph.D.

Academic Editor

PLOS ONE

Journal Requirements:

Additional Editor Comments:

Substantial revisions wrt to the queries raised by both the reviewers.

Reviewers' comments:

Reviewer's Responses to Questions

**Comments to the Author**

1. Is the manuscript technically sound, and do the data support the conclusions?

Reviewer #1: Partly

Reviewer #2: Partly

2. Has the statistical analysis been performed appropriately and rigorously? 

Reviewer #1: No

Reviewer #2: No

3. Have the authors made all data underlying the findings in their manuscript fully available?

Reviewer #1: Yes

Reviewer #2: No

4. Is the manuscript presented in an intelligible fashion and written in standard English?

Reviewer #1: No

Reviewer #2: No

5. Review Comments to the Author

Reviewer #1: This manuscript developed EST-SSR makers for Iron-Heart Cunninghamia lanceolat based on the transcriptome data. The results might provide some information for analyzing the genetic diversity of Chinese fir.

However, there are many problems in this manuscript.

Firstly, the whole languages were very poor, and need to further be reedited and reviewed. Generally, the writing of the manuscript was used the third person. In this paper, the whole was used the first person.

Each section of the manuscript seems mixed and unrefined. In abstract, it seemed long that makes some key results not underlined. Some sentences were repeat used in the introduction and discussion. Some sentences were long and disorderly.

In methods section, the writing of genetic diversity index was nonstandard.

Within Figure 4, the abbreviation should be noted in the title.

Discussion section, there were many sentences repeat with the results. Thus, “Unigene assembly and annotation” and “Marker discovery” could be merged.

In “Transferability of SSR markers ……”, the genetic diversity level of Iron-Heart C. lanceolate far lower than those in Duan Hongjing’s study…., “The phylogenies of the red-heart Chinese fir and Iron-Heart C. lanceolat were relatively close…..” These should be much related the limited samples that make the rationality insufficient.

Reviewer #2: This piece of work demonstrates the versatility of EST-SSR markers for phylogenetic evaluation and genetic diversity analysis of C. lanceolata. Such work should be attempted in future as such markers and the primers developed are vital resources for endemic species evaluation for conservation and beneficial needs. But the presentation and the detailing of the research here needs a lot to be desired. In fact, the research results presented is fragmented and could have been detailed. Please take care of the following points to considerably improve the chances of publication in this journal.

1. English editing of the revision is a must. At most places, clarity is missing and there are insincerities in writing.

2. Title- please edit: De novo assembly of Iron-Heart Cunninghamia lanceolata transcriptome and EST-SSR marker development for genetic diversity analysis

3. Abstract is data heavy and long. Just discuss brief results for a summative understanding.

4. Mononucleotide repeats should not be considered at all. Please focus from di- to hexa-nucleotide repeats.

5. Please include RepeatMasker based repeat analysis (additional data).

5. Please include COG-based annotation results.

6. Why PAGE and silver nitrate solution was used for determination of PCR products? Why not agarose-based determination?

7. Whether the unigenes obtained were all 'coding for ORFs'?

8. How did you differentiate between KEGG and pathway analysis?

9. Please present the pfam annotations as a Table (1902 annotations were unique)

10. Please provide figure 3A and B as a table and present the motif types as a figure.

11. Discussion is weak. Please discuss your data with previous findings.

12. BioProject??

6. PLOS authors have the option to publish the peer review history of their article (what does this mean?). If published, this will include your full peer review and any attached files.

Reviewer #1: No

Reviewer #2: **Yes: **Bharat Bhusan Patnaik

---

## [Author Response · Author response to Decision Letter 0]

10 Jul 2023

Response to Reviewers

Dear PH.D Pramod Prasad,

Thank you for offering us once more an opportunity to resubmit a revised manuscript. Here, we submit the revised manuscript entitled “De novo assembly of Iron-Heart Cunninghamia lanceolata transcriptome and EST-SSR marker development for genetic diversity analysis” (ID: PONE-D-22-35149 ) to PLOS ONE.

We appreciate your letter and the reviewers’ comments concerning our manuscript. These comments are all valuable and very helpful for revising and improving our paper, as well as the important guiding significance to our researches. We have studied comments carefully and have made correction which we hope meet with approval. Revised portion are marked by using 'track changes' in the paper. 

Detailed responses to associate editor and the two reviewers´ comments are provided in the next sections. 

Therefore, I would be greatly appreciated for that you can speed up the review process.We hope you find the improvements to the manuscript satisfactory. Please feel free to contact us with any questions and we are looking forward to your response.

Thank you and best regards.

Yours sincerely,

Can Xiao

E-mail: 17916370@qq.com

---

## [Decision Letter · Decision Letter 1]

16 Aug 2023

PONE-D-22-35149R1De novo  assembly of Iron-Heart Cunninghamia lanceolata  transcriptome and EST-SSR marker development for genetic diversity analysisPLOS ONE

Dear Dr. Xiao,

Thank you for submitting your manuscript to PLOS ONE. After careful consideration, we feel that it has merit but does not fully meet PLOS ONE’s publication criteria as it currently stands. Therefore, we invite you to submit a revised version of the manuscript that addresses the points raised during the review process.

Please make necessary changes suggested by the reviewers.==============================

We look forward to receiving your revised manuscript.

Kind regards,

Pramod Prasad, Ph.D.

Academic Editor

PLOS ONE

Journal Requirements:

Reviewers' comments:

Reviewer's Responses to Questions

**Comments to the Author**

1. If the authors have adequately addressed your comments raised in a previous round of review and you feel that this manuscript is now acceptable for publication, you may indicate that here to bypass the “Comments to the Author” section, enter your conflict of interest statement in the “Confidential to Editor” section, and submit your "Accept" recommendation.

Reviewer #3: All comments have been addressed

2. Is the manuscript technically sound, and do the data support the conclusions?

Reviewer #3: Yes

3. Has the statistical analysis been performed appropriately and rigorously? 

Reviewer #3: Yes

4. Have the authors made all data underlying the findings in their manuscript fully available?

Reviewer #3: (No Response)

5. Is the manuscript presented in an intelligible fashion and written in standard English?

Reviewer #3: Yes

6. Review Comments to the Author

Reviewer #3: PONE-D-22-35149_R1.

Manuscript entitled “De novo assembly of Iron-Heart Cunninghamia lanceolata transcriptome and EST-SSR marker development for genetic diversity analysis” by Liu et al describes transcriptome sequencing and annotation of Iron-Heart C. lanceolata, mined SSRs from transcriptome and development of 15 polymorphic EST SSRs. It adds 15 novel SSR markers to the valuable species having less genomic resources. The experimental design and the approaches used in this work seem both correct for the most part. Overall, the article is informative and written well. However, in my opinion, some aspects of the manuscript need to be revised before considering this work suitable for publication.

Following are few comments based on the R1 copy

1. Line # 125-130: explains the 3 objectives. I feel the third objective is redundant and it forms the part of 2nd objective. This could be modified suitably.

2. Line #340 & 465: Word transferability is used to denote the SSR markers amplifications in accessions from different regions. But the term transferability is used for their amplification or applicability across species i.e., cross species amplifications. So, usage of this term could be avoided here.

3. Table 4: Generally, the SSRs are classified as class I (>20 bp) & ClassII (12-20 bp) based on size of repeat motifs. Which allows the users to select markers. Usually, Class I are more polymorphic. Please add few lines in the results and discussion based on this classification

7. PLOS authors have the option to publish the peer review history of their article (what does this mean?). If published, this will include your full peer review and any attached files.

Reviewer #3: **Yes: **Dr. Siddanna Savadi

---

## [Author Response · Author response to Decision Letter 1]

29 Sep 2023

Point 1: Line # 125-130: explains the 3 objectives. I feel the third objective is redundant and it forms the part of 2nd objective. This could be modified suitably Response: Indeed, we agree with your suggestion very much. After comprehensive consideration, this study mainly has two goals, and we have already made revisions in the line 77-79 in the Revised Manuscript with Track Changes.

Point 2: Line #340 & 465: Word transferability is used to denote the SSR markers amplifications in accessions from different regions. But the term transferability is used for their amplification or applicability across species i.e., cross species amplifications. So, usage of this term could be avoided here.

Response: Thank you very much for your valuable comments, we have replaced word “transferability” with word “cross-species transferability” in line 243 and line 346 in the Revised Manuscript with Track Changes.

Point 3: Table 4: Generally, the SSRs are classified as class I (>20 bp) & ClassII(12-20 bp) based on size of repeat motifs. Which allows the users to select markers. Usually, Class I are more polymorphic. Please add few lines in the results and discussion based on this classification

Response:The characteristically short lengths of SSRs may have functional implications with respect to their evolution or the genes involved in plant physiology and development. We also read the article on the classification of grades according to the length of the SSR, and we made the corresponding result analysis and discussion in our Revised Manuscript with Track Changes in the line 237-240 and line 339-345.

---

## [Editor Report · Decision Letter 2]

10 Oct 2023

De novo  assembly of Iron-Heart Cunninghamia lanceolata  transcriptome and EST-SSR marker development for genetic diversity analysis

PONE-D-22-35149R2

Dear Dr. Xiao,

We’re pleased to inform you that your manuscript has been judged scientifically suitable for publication and will be formally accepted for publication once it meets all outstanding technical requirements.

Kind regards,

Pramod Prasad, Ph.D.

Academic Editor

PLOS ONE

Additional Editor Comments (optional):

The manuscript is now suitable for publication. 
---

## [Editor Report · Acceptance letter]

25 Oct 2023

PONE-D-22-35149R2 

*De novo* assembly of Iron-Heart *Cunninghamia lanceolata* transcriptome and EST-SSR marker development for genetic diversity analysis 

Dear Dr. Xiao:

I'm pleased to inform you that your manuscript has been deemed suitable for publication in PLOS ONE. Congratulations! Your manuscript is now with our production department. 

Kind regards, 

on behalf of

Dr. Pramod Prasad 

Academic Editor

PLOS ONE